# Leveraging insurance customer data to characterize socioeconomic indicators of Swiss municipalities

**Lorenzo Donadio[1], Rossano Schifanella[2,3], Claudia R. Binder[1], Emanuele Massaro[1] ***

**1** ENAC, HERUS Lab, Ecole polytechnique federale de Lausanne, IIE, Lausanne, Switzerland, **2** University of Turin, Turin, Italy, **3** ISI Foundation, Turin, Italy

* ema.massaro@gmail.com

**Data Availability Statement:** Statistical data used in this research are available online at the Swiss federal statistical office website https://www.bfs.admin.ch/. The insurance costumer data can be provided for research purposes by the insurance company "La mobilière" https://www.mobiliere.ch/.

## Abstract

The availability of reliable socioeconomic data is critical for the design of urban policies and the implementation of location-based services; however, often, their temporal and geographical coverage remain scarce. We explore the potential for insurance customers data to predict socioeconomic indicators of Swiss municipalities. First, we define a features space by aggregating at city-level individual customer data along several behavioral and user profile dimensions. Second, we collect official statistics shared by the Swiss authorities on a wide spectrum of categories: Population, Transportation, Work, Space and Territory, Housing, and Economy. Third, we adopt two spatial regression models exploring both global and local geographical dependencies to investigate their predictability. Results show consistently a correlation between insurance customer characteristics and official socioeconomic indexes. Performance fluctuates depending on the category, with values of $R^2 > 0.6$ for several target variables using a 5-fold cross validation. As a case study, we focus on predicting the percentage of the population using public transportation and we discuss the implications on a regional scope. We believe that this methodology can support official statistical offices and it could open up new opportunities for the characterization of socioeconomic traits at highly-granular spatial and temporal scales.

## 1 Introduction

National Statistical Institutes (NSIs) play an important role in modern societies to release precise information on social, environmental or economical activities [1] in the form of a census. The census records key aspects such as the population living in an area, their age, gender, income, and it enables predictive scenarios to estimate the need for schools, residential homes or public services. Censuses are an essential component to estimate the indicators that measure the progress towards the realization of the Sustainable Development Goals [2]. Official statistics on socioeconomic status are increasingly addressing a significant modernization of their production process, nationally and internationally [3]. This is also due to the opportunities offered by the use of new data sources, such as mobile phone data [4], social media [5], satellite

**Funding:** This study was supported by La Mobilière Insurance, Characterization of Quality of Space of Urban System, in the form of funding awarded to the HERUS Lab for the salary of a postdoc for the duration of 2 years and through coupling the data of insurance customers with statistically available data (proposal reference 18219_181217). The funder had no further role in study design, data collection and analysis, decision to publish, or preparation of the manuscript.

**Competing interests:** The authors have read the journal's policy and have the following competing interests: La Mobilière Insurance, Characterization of Quality of Space of Urban System, provided funding to the HERUS Lab for the salary of a postdoc for the duration of 2 years, as well as support through coupling the data of insurance customers with statistically available data. This does not alter our adherence to PLOS ONE policies on sharing data and materials. There are no patents, products in development or marketed products associated with this research to declare.

images [6], credit card transactions [7] and others [8–10]. One of the main challenges for the NSIs is to coherently integrate new and traditional sources of investigation, with an increasingly widespread orientation towards the construction of registers of integrated elementary data. Three important challenges arise: i) the data collection methodology and quality, ii) privacy and legal issues and iii) the processing, storage and transfer of large data sets. Data sources such as social media, and mobile phone records, do not have a well-defined target population, structure and quality (see Section 2 for a literature review) that make difficult to apply traditional statistical methods based on sampling theory. Privacy and legal aspects pose another challenge: the prevention of the disclosure of the identity of individuals is regulated and enforced by international laws, and ensuring an appropriate level of privacy is challenging in case of heterogeneous, and multi-source large scale data streams [11]. Copyright and data ownership [12] provide a barrier to open sharing platforms. Moreover, data processing represents an additional challenge due to the technological difficulties in the storage and the transfer of large amount of heterogeneous information ensuring security [13]. In this context, countries are increasingly favouring alternative means of gathering information, instead of the *traditional* techniques of sending out printed forms, interviewing people in person, or via the use of online questionnaires. Alternatively, they are looking to indirect means of collecting data, taking advantage of a wide spectrum of administrative data streams that act as a proxy for the variables of interest.

In this direction, customers insurance records represent a valuable input to model the socioeconomic substrate of cities, and an opportunity for policy makers and researchers to broaden the scope of their studies. Social scientists raised the issue of representativeness and sampling bias of large scale digital data. For example, in [14] the authors show how age, gender, ethnicity, socioeconomic status, online experiences, and Internet skills, influence the social network sites that users generally adopt. This has implications for the extent of the conclusions that a study could claim given a particular audience. Like census data, insurance customers records share a similar size, reliability, and structural complexity [15]. However, they differ in their spatio-temporal granularity and collection costs. In fact, the information of insurance customers is collected constantly by the provider while the census runs generally with a multi-yearly frequency due to its organizational costs. A downside is the proprietary nature of customers records that could invalidate the possible benefits for a broader community. However, we embrace the vision of initiatives like *Data Collaboratives*: https://datacollaboratives.org. that propose a new form of collaboration, beyond the public-private partnership model, in which participants from different sectors, in particular companies, exchange their data to create public value. In this research, we develop a methodology to predict socioeconomic indicators at a city level using individual customers data from an insurance company in Switzerland. Swiss municipalities, sometimes also called *communities*, are the lowest administrative level in the country. The responsibilities of the 2.212 (as of 1 January 2019) Swiss municipalities is decided by each Cantons. These may include the provision of local public services such as education, medical and social assistance, public transport and tax collection. Their degree of centralization depends on the choice of the single canton. Municipalities are generally governed by a council headed by a mayor (executive power) and by the general assembly of all adult Swiss residents (legislative power). Many cantons leave the larger municipalities the option of opting for a city parliament. Swiss citizenship is based on the citizenship of a municipality. Every Swiss citizen is, first, a citizen of a municipality (right of citizenship of the city or of origin) and, then, of a canton (right of cantonal or indigenous citizenship). For all these reasons, our analysis adopts the municipality as a spatial unit of reference.

We propose a two-steps process to predict a wide range of socioeconomic indicators. First, we compute a set of behavioral metrics using customers activity logs concerning housing

properties and vehicles insured by "La Mobili'ere" in 2017. Second, we use those microeconomics indicators as explanatory variables on two spatial regression models to predict 12 socioeconomic indices of 170 Swiss municipalities for which we have reliable official statistical data as ground truth. In this work, we focus on indices related to six different categories, i.e., Population, Transportation, Work, Space and Territory, Housing, and Economy. We show that insurance data customers can represent a valid resource to model socioeconomic indicators at scale.

The rest of the paper is organized as follows. Section 2 describes how insurance data can benefit the urban data science research agenda and it provides an overview of the previous work in this area. Section 3 illustrates the two data sets that we use in this paper: insurance and census data. In Section 4 we describe the methodology and the modeling framework. Section 6 discusses the results presented in Section 5 and it provides a critical view on the limitations of the implemented approach. Finally, Section 7 summarize the importance and the impact of this research and it provides insights for future directions.

## 2 Related work

Researchers across various disciplines including sociology, demography and public health have been keen on examining how society functions observing populations at scale. Socioeconomic indicators of cities, which were investigated before the digital era, collected data using field studies [16] or surveys [17]. The focus of those studies was mostly finding correlations between demographic factors differences in urban and suburban areas [16], crime rate [18], population health [19], residential segregation [20] or waste production [17]. In all these cases the results were based on the active participation of individuals to surveys, that may have been affected by the tendency of respondents to alter their behavior knowing that they are *monitored*. On the contrary, the digitalization of the modern society allows to model human behavior by means of indirect and less intrusive data collection methodologies, that are often a byproduct of services designed for different purposes. In a scenario where the collectivity produces every day more digital footprints than we are able to process, the majority of the recent studies focused on the use of *big data* streams to predict and study socioeconomic indicators of cities and countries. An extensive body of work adopt digital traces such as cell phone records [4, 21], social media posts [22], vehicle GPS traces [23] or credit card transactions [7, 24] to model human dynamics at scale. For example, it has been shown that social media data can predict the interplay between demographic attributes and gender gap [25], the monitoring [26] and assimilation of migrants [27], unemployment [28] rate, or health outcomes [9]. Social media are increasingly used for demographic attribute prediction [29]. Exploiting user interactions with web search engines, Weber and Castillo [30] inferred gender, income, and ethnicity, while Weber and Jaimes [31] exploited the US census to highlight the differences in navigational and search patterns among several demographic groups. Gender and age are also inferred using call detail records from smartphone devices over a large population.

Our contribution belongs to this line of work, however, to the best of our knowledge, it explores for the first time the use of insurance customer records to predict census variables. Insurance data have been mostly used to study the impact of specific diseases [32, 33], to propose models of customers fraud detection [34, 35], to explore the correlation between census-based socioeconomic indicators and injury causes [36] or to evaluate disparities within health care systems [37].

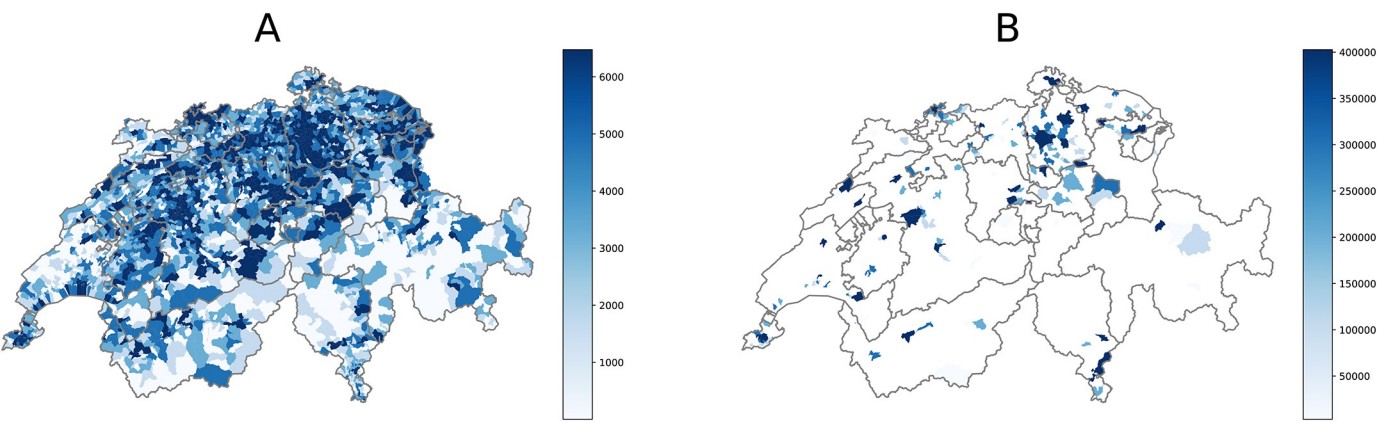

**Fig 1. Comparison between the spatial coverage of the insurance customers and the Union of Swiss Cities datasets.** (A) Insurance customers at zip code level: we count around 1.35M units which represent nearly 15% of the Swiss population. (B) Number of inhabitants per municipality from the Union of Swiss Cities dataset. Note that the geographical coverage of the official statistics is limited to the Swiss municipalities with at least 10K inhabitants covering 9.35% of the country's surface. To ensure a match between the two data sources, we limit the scope of the analysis to this subset of cities.

## 3 Data

In this work, we make use of two data sets: 1) the customer activity logs of a Swiss insurance company and 2) the official socioeconomic indicators of all the Swiss municipalities with more than 10,000 inhabitants. The indicators are collected by the initiative called *Union des villes Suisse*: https://uniondesvilles.ch/ that publishes statistics on 173 Swiss municipalities every year since 2006. In this research, we focus on a cross-sectional snapshot for 2017 [38]. The two datasets have different spatial aggregations: while the information on the insurance customers is at zip code level; the socioeconomic indicators have been collected at municipality level. To match the spatial granularity, we restrict our analysis to the 170 municipalities that are present in both datasets. Fig 1 compares the spatial coverage in the two cases, while Table 1 highlights differences and similarities across several dimensions.

### 3.1 Insurance data

The dataset contains the housing and vehicles insurance policies of 1,341,328 anonymized customers of La Mobiliʻere who were active during 2017. La Mobiliʻere is a Swiss insurance group (brands: Die Mobiliar, La Mobiliʻere, La Mobiliare) that is organized as a holding company headed by a cooperative. The company was founded in 1826, making it the oldest private insurance company in Switzerland. With a market share of over 29%, it is the leader in the personal property insurance market. For each user, we have three classes of information: i) *demographic*, e.g., age, gender, zip code of the residential area, employment and civil status; ii) *cars*,

**Table 1. Comparison between the two datasets.**

|  | Insurance Data | Union of Swiss Cities |
| --- | --- | --- |
| Frequency | Every year | Every year |
| Spatial aggregation | Zip code | Municipality |
| Data points | 3,185 | 173 |
| Cost | Not expensive | Expensive |
| Design | For insurance marketing | For statistical analysis |
| Availability | Private | Public |

**Table 2. Information for each customers in the insurance dataset.**

| Catergory | Variable Name | Description | Variable Type |
|---|---|---|---|
| Demographic | Nmbr | Anonymous ID | Alphanumeric |
| | JobState | Employement status | String |
| | Civil | Civil Status | String |
| | Gender | Gender | String |
| | YearOfBirth | Year of birth | Integer |
| | Own/Rent | If own or rent an house | Boolean (Yes/No) |
| | Lang | Speaking language | String (French, German, Italian) |
| | Nation | Nation of origin | String |
| | ZIP | Zip code of residence | 5-digit code |
| | Children_0-26 | How many children | Integer |
| Cars | Car1_Canton* | Canton where the car is registered | String |
| | Car1_Brand* | Brand of the car | String |
| | Car1_Price* | Price of the car (CHF) | Integer |
| | Car1_ccm* | Cylinder capacity | Integer |
| | Car1_ClaimsCt5Y* | Number of claims over the last 5 years | Integer |
| | Car1_ClaimsSum5Y* | Sum of the money from the claims over the last 5 years (CHF) | Float |
| | Car_Premium | Premium class | String |
| Housing | HH_Zip | Zip Code of the insured house | 5-Digit code |
| | HH_Ins_Sum | Total sum of the insured values of the house (CHF) | Integer |
| | Stand_of_furn | Standard of furniture | Integer (1-2-3-4-5) |
| | Rooms | Number of rooms | Integer |
| | Build_Zip | Zip Code of the insured building | 5-Digit code |
| | Build_Ins_Sum | Total sum of the insured values of the building (CHF) | Integer |
| | Year_ofcontrs | Year of constructions of the building | 4-Digit Integer |
| | Type | Type of building | String |
| | HHaB_ClaimsCt5Y | Number of claims over the last 5 years | Integer |
| | HHaB_ClaimsSum5Y | Sum of the money from the claims over the last 5 years (CHF) | Integer |
| | HH_and_Bld_Prem | Premium class of the building | String |

e.g., how many cars are insured, the brand and the price of the vehicles, as well as the record of the claims and the respective compensations; iii) *housing*, e.g., the number of private buildings or houses insured, the price of the building and the logs of the claims. Customers' details are aggregated at the level of the 170 municipalities for which we have official census data using the zip code as spatial reference; this step leaves us with 568,426 customers matching the geographical boundaries. Table 2 summarizes the information available.

## 3.2 Swiss census data

The official statistics for the Swiss municipalities are collected and made available online within the initiative *Statistics of Swiss Cities* that is is the result of a collaboration between the Union of Swiss Cities and the Federal Statistical Office (FSO) https://www.bfs.admin.ch/l. The report is published in the first quarter of the year and it presents varied facets of the urban life; we focus on six domains: *population* (*p*), *transportation* (*t*), *employment* (*w*), *space and territory* (*s*), *housing* (*h*) and *economy* (*e*). We collect a total of 86 indicators for each municipality: 11 for transportation, 29 for population, 11 for employment, 8 for space and territory, 18 for housing and 9 for economy. From the original dataset, we focus on the key target variables that are not redundant and that can be a proxy for quality of life in cities, such as the

**Table 3. List of the target indicators for the 6 different domains.**

| Domain | Variable |
| --- | --- |
| Population | $p_1$: Fraction of foreigners |
| | $p_2$: Fraction of beneficiaries of social assistance |
| Transportation | $t_1$: Cars per 1000 inhabitants |
| | $t_2$: Fraction of commuters using public transportation |
| Employment | $w_1$: Unemployment rate |
| | $w_2$: Unemployment rate among women |
| Space and Territory | $s_1$: Area covered by buildings (%) |
| | $s_2$: Green area (%) |
| Housing | $h_1$: Vacancy rate (%) |
| | $h_2$: Average area per inhabitant in square meters |
| Economy | $e_1$: Municipal debt |
| | $e_2$: Fraction of investment in culture |

unemployment rate [24], use of public transportation [39] or investment in culture [40]. As a result of this process, we restrict the analysis to two indicators for each domain: the fraction of foreigners and the rate of beneficiaries of social assistance ($p$), the number of private cars per person and the fraction of commuters using public transportation ($t$), the unemployment rate and the unemployment rate among women ($w$), the percentage of areas covered by buildings or green areas ($s$), the vacancy rate and the average area per inhabitant ($h$), and the municipal debt and fraction of investment in culture ($e$). The complete list of selected target variables is summarized in Table 3.

### 3.3 Validation

As a validation step, we test the representativeness of the insurance data along four dimensions: (a) total population, (b) percentage of foreigners, (c) percentage of population aged 20-40, and (d) percentage of population aged 0-19. Fig 2 shows, for each dimension, a scatter plot and the corresponding Pearson's correlation coefficient computed using the official census data and the La Mobili͏ere customers base. A high degree of correlation ($\rho = 0.91$) can be observed for the total population variable, meaning that the insurance dataset mimics quite well the population distribution at the municipality level. Focusing on age, we observe a strong relation for the case of customers in the age range 20-40 ($\rho = 0.8$) while the correlation disappears ($\rho = -0.05$) for customers aged 0-19. This behavior is expected since children and teenagers are not usually the owners of insurance policies on vehicles or houses. Last, we observe a solid relation with the percentage of foreigners ($\rho = 0.6$).

It is worth noting that socioeconomic processes often manifest non-random spatial patterns that make close areas more similar than distant ones. Moreover, spatial effects do not apply only to the case of neighboring areas; on the contrary, a consistent body of literature in geography define spatial relationships between aerial units as a function of distance [41]. Often this choice depends on prior knowledge about the area under study or a conceptualization of the interactions between neighboring locations with regards to the target quantity. In this work, we refer to the Moran's I statistic [42] to assess the presence of spatial autocorrelation in the census variables. Moran's I measures the global spatial autocorrelation of an attribute $y$ measured over $n$ spatial units using the following relation:

$$I = \frac{n}{s_0} \sum_i \sum_j z_i w_{i,j} z_j \frac{1}{\sum_i z_i z_j} \tag{1}$$

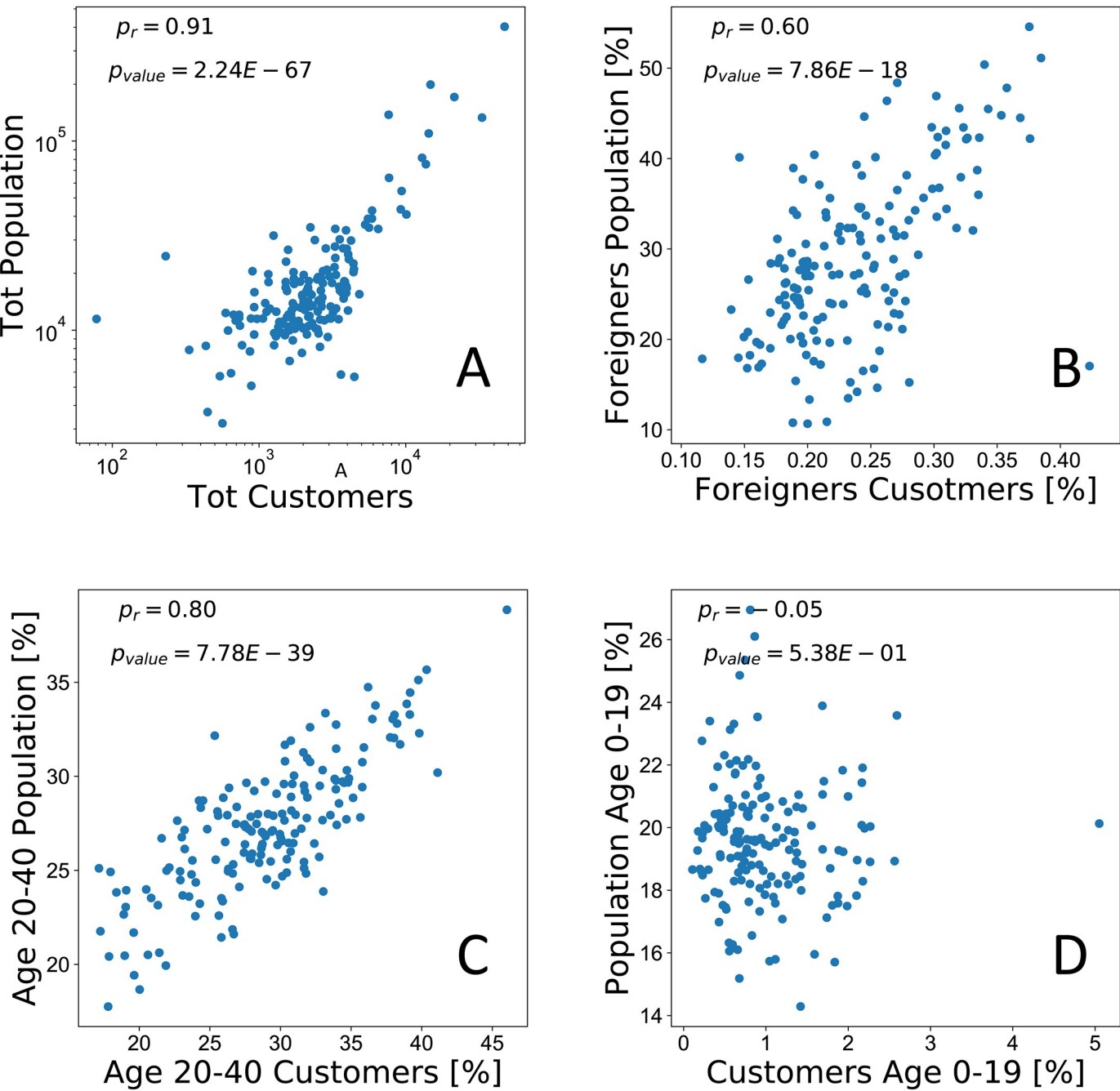

**Fig 2. Pearson's correlation between aggregated information of the insurance customers (x-axis) and the census data (y-axis).** Each point corresponds to a different municipality. On the top, we report the correlation between the total number of inhabitants and customers (A) and between the fraction of foreigners versus the fraction of insured foreigners (B). On the bottom, we show the correlation between the fraction of the population aged between 20-40 years (C) and 0-19 years (D) with the fraction of customers within the same age range.

where $w_{i,j}$ are the spatial weights, $z_i = y_i - \bar{y}$ with $\bar{y}$ being the average across spatial units, and $s_0 = \Sigma_i \Sigma_j \, w_{i,j}$. In our experimental settings, the spatial weights are computed using endogenous adaptive bandwidths with a Gaussian kernel function implemented in the Python package *pysal*: https://pysal.readthedocs.io/en/latest/. Table 4 shows how all the selected target variables

**Table 4. Moran's I coefficients for the main census variables.**

| Variable | Moran's I |
|---|---|
| $p_1$: Fraction of foreigners | 0.7 |
| $p_2$: Fraction of beneficiaries of the social assistance | 0.73 |
| $t_1$: Cars per 1000 inhabitants | 0.56 |
| $t_2$: Fraction of commuters using public transportation | 0.76 |
| $w_1$: Unemployment rate | 0.8 |
| $w_2$: Unemployment rate between women | 0.79 |
| $s_1$: Building area (%) | 0.74 |
| $s_2$: Green area (%) | 0.64 |
| $h_1$: Vacancy rate (%) | 0.66 |
| $h_2$: Average area per inhabitant in square meters | 0.78 |
| $e_1$: Municipal debt | 0.61 |
| $e_2$: Fraction of investment in culture | 0.7 |

are positively spatially autocorrelated, ranging from $I = 0.56$ to $I = 0.8$. This implies that municipalities that are closer in space tend to share similar socioeconomic characteristics.

## 4 Methods

In this section, we describe the methodological steps of our predictive pipeline. After a features selection module, we adopt multivariate linear regression to predict the socioeconomic indicators of interest using two spatially-aware models that capture the global and local geographical dependencies.

### 4.1 Features selection

The first step in constructing cost-effective predictors is to select the features that will best predict a given outcome variable. For each of the socioeconomic indicators in Table 3, we select a subset of explanatory variables from the initial pool of covariates summarized in Table 5 using the *LassoLarsIC* module available in scikit-learn *see* https://scikit-learn.org. To reduce model complexity and to prevent overfitting, *LassoLarsIC* adopts the Least Absolute Shrinkage and Selection Operator [43] (LASSO) model for fit and it relies on the *Least Angle Regression* [44] (LARS) and the *Bayes Information Criterion* [45] (BIC) for model selection, trying to find the right trade-off between fitting performance and the complexity of the model. We explore alternative regressors with variable section, in particular the *Elastic Net: see* https://scikit-learn.org approach that exploits an iterative fitting procedure along a regularization path, and the *Multitask LASSO: see* https://scikit-learn.org method that adopts L1/L2 mixed-norm as regularizer. These approaches tend to select a wider range of variables with a minimal increase to the end-to-end performance for some of the target variables. In the final experimental pipeline, we adopt the *LassoLarsIC* implementation due to its ability to strike the right trade-off across the use cases under study.

note https://scikit-learn.org/stable/modules/generated/sklearn.linear_model.LassoLarsIC.html that relies on Least Angle Regression and the Bayes Information Criterion for model selection and to find a trade-off between the goodness of fit and the complexity of the model.

Since variable selection methods may suffer from model instability or potential bias in parameter estimates and confidence intervals (especially relevant in explanatory modeling), we implemented the methodology and practical suggestions proposed in [46, 47] to control for these effects. In particular, we aim at estimating the stability of the selection procedure to

**Table 5. Final set of features aggregated at the municipality level.**

| Category | Name | Description |
|---|---|---|
| Demographic | f1: | Unemployment rate |
| | f2: | Average age in the municipality |
| | f3: | Fraction of owners (house) |
| | f4: | Fraction of foreigners |
| | f5: | Average number of customers with at least one child |
| | f6: | Market share |
| | f7: | Fraction of women |
| | f8: | Number of customers divided by total customers |
| Cars | f9: | Average price of the cars |
| | f10: | 95th percentile price of the cars |
| | f11: | Average year of the car |
| | f12: | 5th percentile year of the car |
| | f13: | Average CCM of the car |
| | f14: | 95th percentile CCM of the car |
| | f15: | Average number of claims per cars |
| | f16: | 95th percentile number of claims of the car |
| | f17: | Average sum of claims of the car |
| | f18: | 95th percentile number of price of the car |
| | f19: | Average premium of the car |
| | f20: | Percent of insured cars |
| Housing | f21: | Average class of furniture |
| | f22: | 95th percentile class of furniture |
| | f23: | Average number of rooms |
| | f24: | 95th percentile number of rooms |
| | f25: | Average building insured sum |
| | f26: | 95th building insured sum |
| | f27: | Average building year of Construction |
| | f28: | 5th percentile building year of construction |
| | f29: | Average type of building |
| | f30: | Average number of claims per building |
| | f31: | Average sum of claims per building |
| | f32: | 95th sum of claims per building |
| | f33: | Average Insured Premium |
| | f34: | 95th sum of insured premium |

random perturbations of training samples. We implemented a subsampling without replacement routine that randomly selects 80% of the initial datasets and run the selection procedure on the subsample. The subsampling technique has been extensively studied showing its asymptotic consistency even in cases where the classical bootstrap fails [48]. We performed 500 subsampling iterations and we computed the stability estimator proposed by Nogueira et al. [49] that is a frequency-based statistics, ranging 0 to 1 and monotonically increasing as the stability of the feature selection grows. The idea is that the stability measure is a linear function of the sample variances with a strictly negative slope. According to the proposed framework, stability values above 0.75 represent an excellent agreement of the feature sets beyond chance, between 0.75 and 0.4 intermediate to good agreement, while values below 0.4 represent a poor agreement.

## 4.2 Spatial Lag Model

To characterize the influence of neighboring spatial units, we implement a Spatial Lag Model [50] (SLM) where the local effects are encoded adding a term that contains a spatially lagged version of the dependent variable. SLM is an instance of Spatial Autoregressive Models where the additional term is treated as an endogenous variable. More formally, this can be expressed in matrix notation, as follows:

$$y = \alpha + \beta X + \lambda W y + \epsilon \tag{2}$$

where $y$ is the vector of observations on the dependent variable, $X$ is the matrix of observations on the exogenous variables, $W$ is the spatial weighting matrix of known constants, $\beta$ is the vector of regression parameters and $\lambda$ is the scalar autoregressive parameter. The variable $Wy$ is typically known as the spatial lag of $y$.

In literature, different approaches have been adopted to find the most appropriate spatial econometric model specification given an empirical use case. In this work, following the discussion in [51], we use a theoretical rather than data-driven approach to guide the decision. In our scenario, we hypothesize the presence of global endogenous effects where the indirect influence of a spatial unit falls on the entire set of locations, producing high-order effects observable even in spatial units that are not directly connected. We adopt the Spatial Lag Model over different specifications such as the Spatial Error Model, to take into account this dimension. For completeness, we also run the Lagrange Multiplier [52] (LM) diagnostics based on the OLS residuals in their standard and robust forms that are at the core of the methodology described in [53]. Consistently, the SLM specification was selected for two thirds of the dependent variables in Table 3 as the most likely to account for the observed spatial effects.

Focusing on the weighting scheme, we adopt a k-nearest neighbors (KNN) approach in which each spatial unit is connected to a fixed number of $k$ closest neighbors; the hyperparameter $k$ is optimized via a grid search to find the optimal value that maximizes the average $R^2$ across all the target variables. We experiment with alternative distance-based weighting schemes that adopt kernel functions with adaptive bandwidth obtaining comparable results. We use the PySAL package *spreg* to implement the SLM in our experimental settings.

## 4.3 Geographical Weighted Regression

Geographical Weighted Regression (GWR) is a local form of linear regression used to model spatially varying relationships. In regression analysis, the strength and direction of association is indicated by the regression coefficients, with one coefficient given for each variable in the dataset. In GWR, instead of one global coefficient for each variable, coefficients are able to vary according to space. This spatial variation in coefficients can reveal interesting patterns which otherwise would be masked. The general formula for a GWR is an extension of Eq 3 where one regression is calculated for each point using spatial weights.

$$y_i = \beta_{0i} + \sum_{j=1}^{m} \beta_{i,j} X_{i,j} + \epsilon_i. \tag{3}$$

The index $i$ indicates the location of the city of interest. GWR basically fits a set of $\beta$ coefficients for each location:

$$\beta_i = (X^T W_i X)^{-1} X^T W_i y, \tag{4}$$

where $W_i$ is the diagonal matrix of the spatial weights, unique to location $i$. There are various schemes for calculating the weights, nearest neighbors, cubic or exponential kernels. The

weights are computed by the following:

$$w_{i,j} = \exp(-0.5\frac{d_{i,j}^2}{b}),$$ (5)

where $d_{ij}$ is the Euclidean distance between municipalities $i$ and $j$ and $b$ is the bandwidth of the kernel that has to be chosen. For each municipality we calculate a vector of weights and then regress using the final formula (whether linear or with interactions), in order to estimate all indicators for each municipality. The parameters forming the GWR will be the focus of the analysis because of the non-stationarity of the problem. It is interesting to explore how the influence of certain explanatory variables changes from city to city and whether there are underlying tendencies. We estimate a bandwidth for each city and model. The optimal bandwidth is estimated by minimizing an information criterion; in practice, we adopt a corrected version of the AIC that, in contrast with the original definition, is a function of sample size [54]. In more details, in a GWR model with a bandwidth $b$, the $AIC_c$ is given by:

$$AIC_c = 2n\ln\hat{\sigma} + n\ln 2\pi + n\frac{n + tr(S)}{n - 2 - tr(S)},$$ (6)

where $\hat{\sigma}$ is the estimated standard deviation of the residuals; $n$ is the number of observations and $tr(S)$ is the trace of the hat matrix $S$. The hat matrix is the projection matrix from the observed $y$ to the fitted values $\hat{y}$ [55]. Each row of the hat matrix is calculated as follows:

$$\text{row}_i = X_i(X^T W_i X)^{-1} X^T W_i$$ (7)

## 4.4 Evaluation

To test the performance of the predictive pipeline we refer to an out-of-sample validation where the estimation of the regression parameters and the hyper-parameter tuning are performed on a training set using a 80% split and cross-validation, while the predictive performance is tested on the hold-out data (remaining 20%). To cope with the heterogeneity of the population distribution in our sample and to allow to train and test the models with a sample that is representative of the entire spectrum of population size, we implement a stratification approach. It is worth noting that using a random sampling strategy instead, we could end up in the cross validation procedure with splits that contain only highly or low populated municipalities, introducing a bias in the evaluation pipeline. In this direction, we partition the municipalities in three classes: communes with less than 25k inhabitants, those between 25k and 100k, and those with a population higher than 100k. Note that there are five cities that belong to the last class: Zurich, Geneve, Basel, Lausanne and Bern, which represent the five main Swiss cities. In each round of the cross-validation, the procedure ensures that each fold is a fair representation of the whole distribution balancing the three classes. We adopt a 5-cross validation accordingly. In the results section, we report the average and the standard deviation of the models performance due to cross-validation.

## 5 Results

In the first part of this section, we present the results of the features selection process for each of the target indicators. After applying the *LassoLarsIC* method the number of selected features spans from 2 (for the Housing indicator $h_1$) to 9 (for the Economy variable $e_2$). In the column *Features* of Tables 6 and 7 we present the selected features for each model. Overall, we observe a fair degree of robustness to random perturbations with the measure of stability that varies

**Table 6. Summary of the results of the Spatial Lag Model (SLM) for the target indicators in the domains Population, Transportation, and Employment.**

| Variable | Features | Coefficient | Probability | [0.025 | 0.975] |
|---|---|---|---|---|---|
| p1<br>Fraction of foreigners | Intercept | -0.003 | 9.48E-01 | -0.103 | 0.097 |
| | f3 | -0.119 | 1.24E-01 | -0.271 | 0.034 |
| | **f4** | **0.460** | **1.65E-12** | **0.332** | **0.589** |
| | **f7** | **0.265** | **1.06E-04** | **0.130** | **0.400** |
| | f23 | -0.109 | 1.74E-01 | -0.267 | 0.049 |
| | W_dep_var | 0.019 | 4.24E-01 | -0.028 | 0.067 |
| p2<br>Fraction of beneficiaries of social assistance | Intercept | -0.009 | 8.49E-01 | -0.101 | 0.083 |
| | **f1** | **0.298** | **1.23E-07** | **0.187** | **0.409** |
| | **f9** | **-0.417** | **1.39E-02** | **-0.752** | **-0.082** |
| | f13 | 0.134 | 4.45E-01 | -0.213 | 0.482 |
| | **f15** | **0.157** | **1.51E-02** | **0.029** | **0.284** |
| | **f23** | **-0.327** | **6.13E-10** | **-0.431** | **-0.223** |
| | W_dep_var | 0.032 | 1.49E-01 | -0.012 | 0.076 |
| t1<br>Cars per 1000 inhabitants | Intercept | 0.008 | 8.84E-01 | -0.101 | 0.117 |
| | **f1** | **-0.137** | **2.94E-02** | **-0.260** | **-0.013** |
| | **f6** | **0.381** | **1.48E-09** | **0.256** | **0.505** |
| | f7 | -0.119 | 1.08E-01 | -0.264 | 0.027 |
| | f8 | -0.130 | 4.98E-02 | -0.261 | 0.001 |
| | f17 | 0.076 | 2.95E-01 | -0.068 | 0.221 |
| | **f19** | **0.165** | **1.69E-02** | **0.029** | **0.301** |
| | f20 | 0.068 | 3.69E-01 | -0.082 | 0.217 |
| | **f21** | **-0.138** | **4.38E-02** | **-0.272** | **-0.003** |
| | f23 | 0.106 | 1.39E-01 | -0.036 | 0.248 |
| | **f27** | **0.154** | **1.47E-02** | **0.029** | **0.279** |
| | W_dep_var | 0.024 | 3.50E-01 | -0.027 | 0.075 |
| t2<br>Fraction of commuters using public transportation | Intercept | -0.097 | 2.04E-02 | -0.180 | -0.014 |
| | **f3** | **-0.239** | **3.13E-06** | **-0.340** | **-0.138** |
| | **f6** | **-0.101** | **1.63E-02** | **-0.184** | **-0.018** |
| | **f20** | **-0.261** | **1.39E-07** | **-0.358** | **-0.163** |
| | **f22** | **0.120** | **1.39E-02** | **0.024** | **0.217** |
| | **f25** | **0.091** | **3.64E-02** | **0.005** | **0.176** |
| | W_dep_var | 0.119 | 3.23E-16 | 0.090 | 0.148 |
| w1<br>Unemployment rate | Intercept | -0.008 | 8.62E-01 | -0.097 | 0.082 |
| | **f1** | **0.220** | **1.26E-04** | **0.106** | **0.333** |
| | **f4** | **0.187** | **6.07E-04** | **0.079** | **0.295** |
| | **f7** | **0.206** | **1.18E-03** | **0.081** | **0.332** |
| | **f13** | **-0.125** | **1.25E-02** | **-0.225** | **-0.026** |
| | **f23** | **-0.212** | **1.36E-04** | **-0.322** | **-0.102** |
| | W_dep_var | 0.054 | 1.26E-02 | 0.011 | 0.096 |
| w2<br>Unemployment rate between women | Intercept | -0.012 | 7.74E-01 | -0.096 | 0.071 |
| | **f1** | **0.151** | **4.60E-03** | **0.046** | **0.255** |
| | **f4** | **0.150** | **3.05E-03** | **0.050** | **0.251** |
| | f6 | -0.087 | 6.45E-02 | -0.181 | 0.006 |
| | f7 | 0.106 | 7.38E-02 | -0.011 | 0.224 |
| | **f9** | **-0.250** | **1.39E-04** | **-0.379** | **-0.120** |
| | f16 | 0.033 | 5.13E-01 | -0.066 | 0.132 |
| | **f19** | **0.286** | **6.83E-05** | **0.144** | **0.428** |
| | f23 | -0.124 | 5.06E-02 | -0.250 | 0.001 |
| | **f33** | **-0.268** | **5.80E-06** | **-0.384** | **-0.151** |
| | W_dep_var | 0.061 | 2.60E-03 | 0.021 | 0.101 |

**Table 7. Summary of the results of the Spatial Lag Model (SLM) for the target indicators in the domains Space and Territory, Housing, and Economy.**

| Variable | Features | Coefficient | Probability | [0.025 | 0.975] |
|---|---|---|---|---|---|
| s1<br>Building area (%) | Intercept | -0.032 | 5.94E-01 | -0.150 | 0.086 |
| | **f3** | **-0.286** | **2.29E-04** | **-0.439** | **-0.133** |
| | f4 | 0.075 | 2.88E-01 | -0.064 | 0.213 |
| | **f7** | **0.138** | **4.31E-02** | **0.003** | **0.274** |
| | f25 | 0.100 | 1.16E-01 | -0.026 | 0.226 |
| | f31 | 0.134 | 7.74E-02 | -0.016 | 0.283 |
| | W_dep_var | 0.070 | 2.53E-02 | 0.008 | 0.132 |
| s2<br>Green area (%) | Intercept | -0.048 | 4.04E-01 | -0.163 | 0.066 |
| | **f3** | **-0.270** | **2.76E-03** | **-0.448** | **-0.092** |
| | f7 | 0.097 | 1.57E-01 | -0.038 | 0.232 |
| | **f11** | **-0.149** | **1.15E-02** | **-0.265** | **-0.032** |
| | f23 | -0.159 | 7.61E-02 | -0.336 | 0.018 |
| | W_dep_var | 0.088 | 6.98E-05 | 0.044 | 0.132 |
| h1<br>Vacancy rate (%) | Intercept | 0.039 | 5.54E-01 | -0.091 | 0.169 |
| | f3 | 0.043 | 6.29E-01 | -0.132 | 0.217 |
| | **f20** | **0.167** | **2.54E-02** | **0.019** | **0.314** |
| | W_dep_var | 0.157 | 1.19E-04 | 0.076 | 0.237 |
| h2<br>Average area per inhabitant in square meters | Intercept | 0.016 | 7.03E-01 | -0.067 | 0.099 |
| | **f1** | **-0.161** | **1.66E-03** | **-0.263** | **-0.060** |
| | **f2** | **0.142** | **5.85E-03** | **0.040** | **0.244** |
| | **f3** | **0.174** | **8.51E-03** | **0.043** | **0.305** |
| | f4 | -0.030 | 5.68E-01 | -0.133 | 0.073 |
| | **f6** | **0.118** | **1.06E-02** | **0.027** | **0.210** |
| | **f21** | **0.137** | **2.44E-02** | **0.017** | **0.256** |
| | f22 | 0.052 | 3.70E-01 | -0.062 | 0.165 |
| | **f23** | **0.313** | **3.30E-06** | **0.180** | **0.446** |
| | f27 | -0.082 | 1.07E-01 | -0.182 | 0.018 |
| e1<br>Municipal debt | W_dep_var | 0.063 | 1.14E-04 | 0.031 | 0.096 |
| | Intercept | -0.014 | 8.31E-01 | -0.143 | 0.115 |
| | f9 | -0.106 | 2.04E-01 | -0.270 | 0.059 |
| | f16 | 0.098 | 1.84E-01 | -0.047 | 0.243 |
| | **f27** | **0.236** | **4.07E-04** | **0.104** | **0.368** |
| | W_dep_var | 0.081 | 1.03E-01 | -0.017 | 0.180 |
| e2<br>Fraction of investment in culture | Intercept | -0.012 | 8.23E-01 | -0.114 | 0.091 |
| | f1 | 0.122 | 9.03E-02 | -0.020 | 0.265 |
| | f4 | 0.002 | 9.72E-01 | -0.138 | 0.143 |
| | f7 | 0.000 | 9.98E-01 | -0.144 | 0.144 |
| | **f11** | **-0.209** | **1.59E-02** | **-0.380** | **-0.038** |
| | **f12** | **0.029** | **7.49E-01** | **-0.148** | **0.205** |
| | **f21** | **-0.100** | **1.20E-01** | **-0.228** | **0.027** |
| | **f23** | **-0.128** | **7.58E-02** | **-0.271** | **0.014** |
| | **f26** | **0.107** | **9.72E-02** | **-0.020** | **0.234** |
| | **f34** | **0.036** | **6.35E-01** | **-0.113** | **0.184** |
| | **W_dep_var** | **0.131** | **8.81E-09** | **0.086** | **0.176** |

across dimensions. In particular, $p_1$ (0.82) and $t_2$ (0.77) show the highest stability that reaches an excellent level of agreement; $w_2$, $h_2$, $w_1$, $p_2$, $e_2$, $s_2$, $t_1$, $s_1$, and $h_1$ cover a range between good (0.72) and intermediate (0.43) stability (variables are listed in decreasing order), while $e_1$ (0.22) shows a poor agreement. This low value indicates how the model characterizing the municipal debt $e_1$ is highly dependent on variations of the training set to define significant determinants. Consistently, $e_1$ is also the indicator with the lowest performance in the predictive task, indicating how the insurance data is not really able to capture its behavior.

Switching the focus on the predictive task, Tables 6 and 7 summarize the results of the Spatial Lag Model for all the indicators. We present the direction and the intensity of the relations along with confidence intervals; significant determinants are marked in bold.

In the Population domain, the fraction of foreigners ($p_1$) is positively ($\beta = 0.46$) linked to the demographic feature $f_4$ that represents the fraction of foreigners customers of La Mobili'ere and, in the same direction, to the fraction of women $f_7$ ($\beta = 0.265$). Moreover, the percentage of people that receive social assistance ($p_2$) is positively linked to the unemployment rate $f_1$ ($\beta = 0.298$), and the average number of claims per car $f_{15}$ ($\beta = 0.157$). We observe a negative relation with average price of the cars $f_9$ ($\beta = -0.417$) and the average number of rooms $f_{23}$ ($\beta = -0.327$) as indirect proxy for the social class.

In the Transportation domain, the number of cars per 1000 inhabitants $t_1$ shows a negative relation with the unemployment rate $f_1$ ($\beta = -0.137$) and the average class of furniture $f_{21}$ ($\beta = -0.138$). A positive link is found with the market share $f_6$ ($\beta = 0.381$), the average premium of the cars $f_{19}$ ($\beta = 0.165$) and the average years of construction $f_{27}$ ($\beta = 0.154$). Concerning the commuters that use public transportation $t_2$ we observe a negative link with the market share $f_6$ ($\beta = -0.101$) and the fraction of house owners $f_3$ ($\beta = -0.239$). This could be explained by the observation that individuals living in rental houses show a higher frequency of ride-sharing use and commuting using public transportation than those who own their houses [56]. A positive relation is found with the percent of insured cars $f_{20}$ ($\beta = 0.261$), the 95th percentile of the class of the insured furniture $f_{22}$ ($\beta = 0.120$), and the average insured sum per building $f_{25}$ ($\beta = 0.091$).

Focusing on the Work category, the unemployment rate $w_1$ is positively connected to a set of demographics features, primarily the fraction of foreigners $f_4$ ($\beta = 0.187$), the fraction of women $f_7$ ($\beta = 0.206$) and, as expected, the unemployment rate of the La Mobili'ere customers $f_1$ ($\beta = 0.220$). We observe an opposite relation with the average CCM of the cars $f_{13}$ ($\beta = -0.125$), and the average number of rooms $f_{23}$ ($\beta = -0.212$). For the case of the women unemployment rate $w_2$, the dominant features are related to the economic characteristics of the items insured, being the average premium of the cars $f_{19}$ ($\beta = 0.286$) in a positive relation and the average price of the cars $f_9$ ($\beta = -0.250$) or average insured premium $f_{33}$ ($\beta = -0.268$) linked negatively. These observations tend to indicate gender differences in the insurance sector. The fraction of foreigners $f_4$ ($\beta = 0.150$) and the customers unemployment rate $f_1$ ($\beta = 0.151$) behave accordingly to the previous case.

Within the Space and Territory category, both the variables percentage of building area $s_1$ and percentage of green area $s_2$ are negatively connected to the fraction of house owner $f_3$ ($\beta = -0.286$ and $\beta = -0.270$ respectively).

In the Housing domain, the vacancy rate $h_1$ appears to be positively related to the percentage of insured cars $f_{20}$ ($\beta = 0.167$), while the average area per inhabitant $h_2$ shows several positive links to the average age $f_2$ ($\beta = 0.142$), the fraction of house owners $f_3$ ($\beta = 0.174$), the market share $f_6$ ($\beta = 0.118$), the average class of furniture $f_{21}$ ($\beta = 0.137$) and the average number of rooms $f_{23}$ ($\beta = 0.313$). Higher values for $h_2$ corresponds to lower unemployment rate $f_1$ ($\beta = -0.161$).

Finally, in the Economy category the municipal debt $e_1$ is positively related to the average year of constructions of the buildings $f_{27}$ ($\beta = 0.236$) that is in accordance with the literature where modern buildings have been considered a proxy for economic status [57]. Moreover, the fraction of investment in culture $e_2$ is negatively connected to the average year of the car $f_{11}$ ($\beta = -0.209$).

It is worth noting that for a group of indicators, the corresponding predictive models identify significant relations with expected determinants: this is the case of the pair ($p_1, f_4$) where the fraction of foreigners is explained using the information on the nationality of La Mobili'ere customers. A similar case happen for the pairs ($p_2, f_1$), ($w_1, f_1$), and ($w_2, f_1$). However, we think that these not surprising relations do not undermine the validity of the experimental settings for several reasons. First, the considered models identify alternative predictors that are complementary and cross-domain to the target indicators. Second, the observation that a variable constructed from a sample of customers of an insurance company is able to predict a census indicator at the national level is not trivial. This represents another suggestion of the validity of the data collected as a proxy for socioeconomic status. Third, to quantitatively evaluate the impact of these not surprising variables, we compare the performance of the original models with a variation where we remove them. The accuracy in terms of $R^2$ for both the SLM and GWR models remains substantially stable for all the indicators, with an average penalty of 0.034 and 0.014 for SLM and GWR, respectively. Refer to Fig A.13 in S1 Appendix for a detailed comparison.

After the analysis of determinants, we focus on comparing the performance of the global (SLM) and local (GWR) spatial models to a standard multivariate linear regressor (OLS) to quantify to benefit of exploiting spatial relations. We measure the performance using the coefficient of determination $R^2$. As shown in Fig 3(A), both the spatial models outperform OLS across target indicators, with a gain in performance up to 30%. It is worth noting that GWR is able to achieve satisfactory results across categories with $R^2$ values ranging from 0.49 for $s_2$ to 0.83 in the case of $w_1$ or $h_2$. This provides a hint on the potential of insurance customers data to characterize socioeconomic processes embedded in space. GWR is useful as an exploratory technique, as it allows the relationships between independent and dependent variables to vary locally and thus captures contextual factors; however, its usefulness as a prediction tool is debated when it comes to model generalizability. To test the ability to perform out-of-sample predictions we turn to stratified cross-validation as described in Section 4.4. As shown in Fig 3 (B) and 3(C), we observe a decrease of the overall performance; however, especially for certain target variables, we are still able to achieve a reasonable performance on the validation set, for example, $h_2 = 0.6$, $w_1 = 0.53$, $p_2 = 0.49$, and $t_2 = 0.49$. The values of the performances of the models are also reported in the Tables A.1 and A.2 in S2 Appendix.

At last, to shed light on the contribution of the insurance datasets, we refer to a set of baseline models in which each target indicator is predicted using the variables of the other categories from the census data For instance, let us model the fraction of foreigners $p_1$ using the explanatory variables $t_1, t_2, \ldots, e_2$ from Table 3. In Fig 4 we report a comparison between the performance of the census-based baseline and the insurance-based models for the cases of SML and GWR. We observe overall a comparable performance using our approach, with the baseline having a positive delta of 0.019 on average across indicators. This is expected because the baseline is based on official census data where cross-correlation effects are present. However, it is worth nothing that in our reference scenario the census is not available and, as such, the baseline approach not feasible. The observation that insurance customers records are able to achieve comparable results is an additional proof of the potential of this approach.

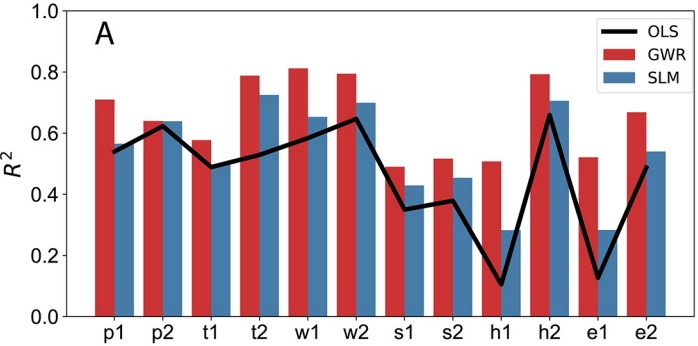

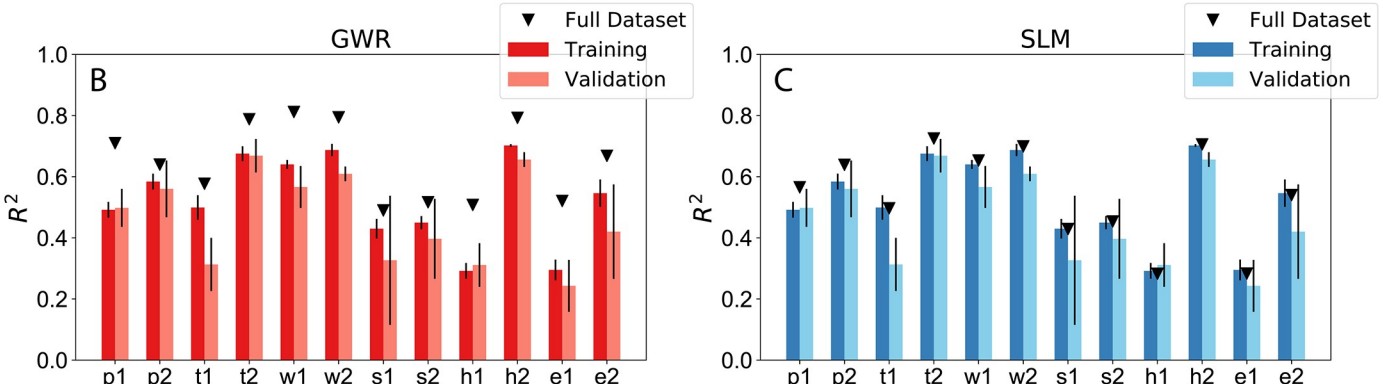

**Fig 3.** (A). Comparison of the performance between the spatial regression models (GWR and SLM) and standard multivariate linear regression (OLS). (B-C) Performance using stratified cross-validation for the full (black triangles), the training and the validation sets respectively. (B) Geographical Weighted Regression and (C) Spatial Lag Model.

## 6 Discussion

In the first part of the paper, we showed how to predict a wide range of socioeconomic indicators using insurance customers activity logs. In this section we shift the attention to a specific use case that has a strong impact on urban mobility and citizens well being: the relation

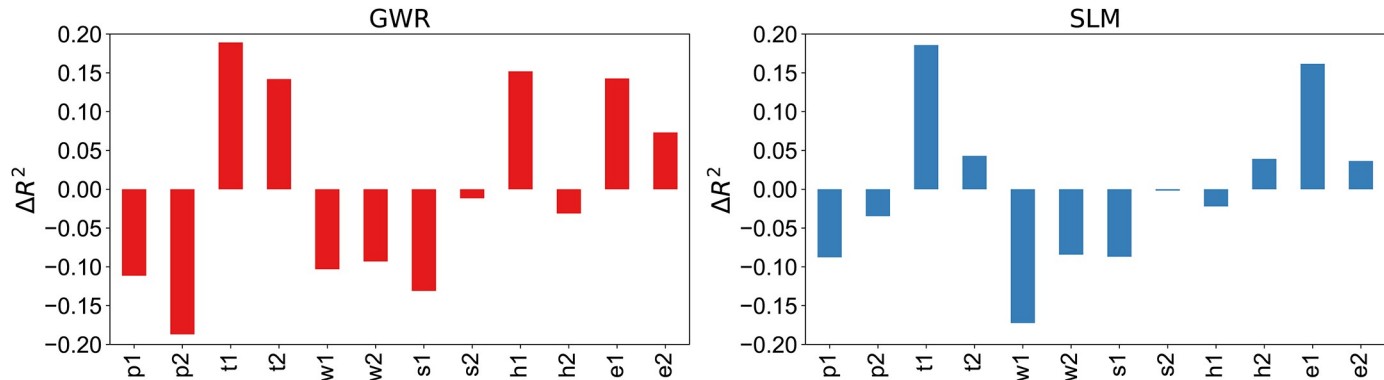

**Fig 4. Comparison between the census-based and the insurance-based explanatory models for the SML and GWR cases.** Positive and negative values mean, respectively, an increase or decrease in performance using La Mobili'ere data in comparison to the census baseline.

**Table 8. Summary statistics for the GWR parameters for predicting $t_2$.**

| Variable | Mean | STD | Min | Max |
|---|---|---|---|---|
| Intercept | 0.005 | 0.313 | -1.169 | 0.339 |
| f3: Fraction of owners (house) | -0.306 | 0.156 | -0.649 | 0.007 |
| f6: Market Share | -0.236 | 0.218 | -0.896 | 0.202 |
| f20: Percent of insured cars | -0.301 | 0.201 | -0.586 | 0.541 |
| f22: 95th percentile class of furniture | 0.197 | 0.131 | -0.019 | 0.58 |
| f25: Average Building Insured Sum | 0.274 | 0.115 | -0.325 | 0.407 |

between commuting and public transport (the variable $t_2$ in our settings). The use of public transportation is an important contributing factor to urban sustainability; it has a heavy environmental footprint reducing air pollution and traffic congestion, among the others. It has also positive financial benefits for families and communities as a whole, higher level of security and direct positive effects on well-being and healthier habits. We chose transportation to exemplify our data analysis as it is the third most important contributor to greenhouse gas (GHG) emissions in the European Union [58].

In Switzerland, transportation accounts for around 31% of the overall GHG emissions [59]. As such, the question of which variables are able to predict the use of public transport is a key issue. In Table 6 and in Table 8 we report the values of the predictors for the global and the GWR models respectively (an analysis of the GWR statistics for the target variables is reported in the Figs A.1 to A.12 in S1 Appendix and in Tables A.3 and A.4 in S2 Appendix). While for the global model, the estimates of the parameters are the same across municipalities, in the GWR case, each municipality has its own local parameters. Accordingly, we report average, standard deviation, minimum, and maximum values as summary statistics. For the GWR, we observe a high variability in the intercept, this is mainly due to the high spatial autocorrelation. The GWR adapts the intercept so it is closer to its neighbors, and thus achieves higher accuracy. More detailed diagnostic information on the regression, such as the kernel bandwidth is provided in Table 9. Turning the attention to the coefficients, we observe that the fraction of customers that own a house ($f_3$) is negatively correlated with the target variable: as expected, it has been observed that individuals living in rental houses show a higher frequency of ride

**Table 9. Information on the GWR of $t_2$; percentage of commuters using public transport.**

| Diagnostic Information | |
|---|---|
| Spatial kernel: | Fixed Gaussian |
| Bandwidth used: | 29.030 |
| Residual sum of squares: | 36.026 |
| Effective number of parameters (trace(S)): | 41.952 |
| Degree of freedom (n—trace(S)): | 128.048 |
| Sigma estimate: | 0.53 |
| Log-likelihood: | -109.337 |
| AIC: | 304.578 |
| AICc: | 334.533 |
| BIC: | 439.268 |
| R2: | 0.788 |
| Adj. alpha (95%): | 0.007 |
| Adj. critical t value (95%): | 2.723 |

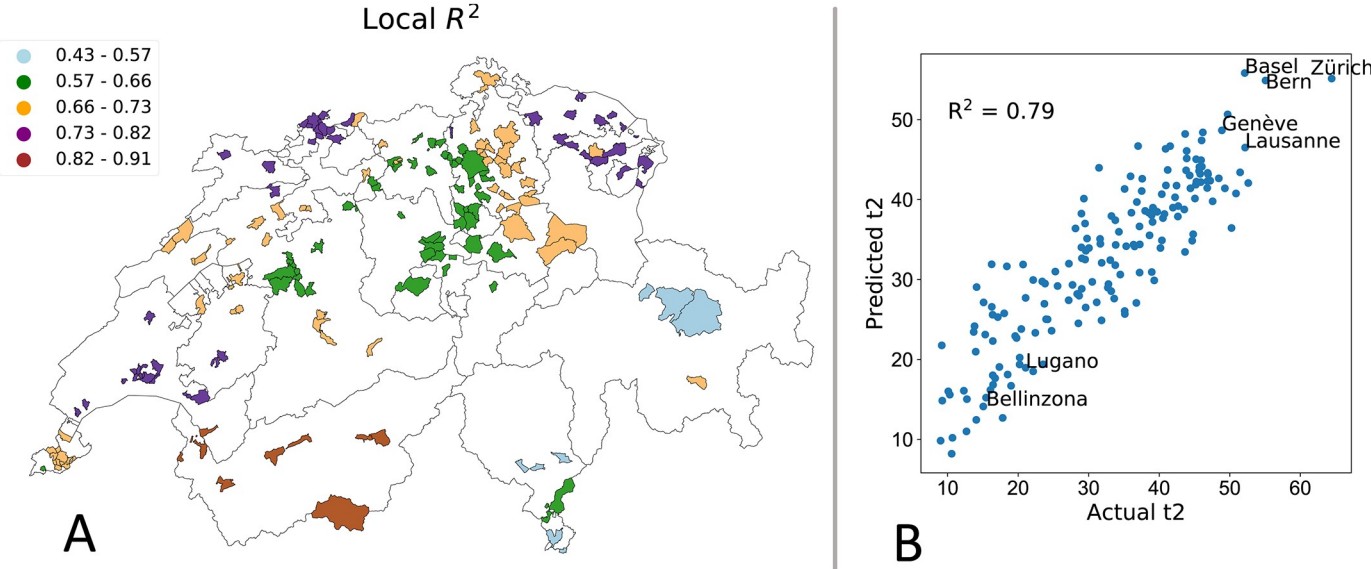

**Fig 5.** (A) Spatial distribution of the local coefficient of determination $R^2$ using GWR to predict the fraction of commuters using public transportation $t_2$. (B) Comparison between predicted and actual values of the percentage of commuters using public transport.

sharing and public transportation adoption than house owners [56]. As expected, also the percentage of insured cars ($f_{20}$) is negative linked to the probability of commuting via public transport; the more cars an individual possesses the less she turns on the public system when it comes to mobility. Moreover, we observed a higher public transport adoption in major cities, e.g., Zurich, Basel, Bern and Geneva as shown in Fig 5(b). This is consistent with our analysis, in fact, the fraction of house owners ($f_3$) is lower in major cities, where prices are higher and people tend not to settle and start a family. Another variable that seems to be significative is the market share ($f_6$). We believe that the market share feature is representative information because even if, the insurance company La Mobili'ere is a national company, is not used equally across the Swiss country because of the competition between different insurance companies. Moreover, having an insurance contract is mandatory also for renting an apartment, the information of the market share of a given company can tell us important information about a certain kind of population living in that area. One of the main characteristic of GWR is that the inferred relationships vary by locality, that implies each municipality has a different fitting performance $R^2$ and coefficients. In Fig 5 we show spatial distribution of GWR accuracy in different regions. Mapping the local $R^2$ values could provide a useful tool to identify areas where the independent variables might or might not explain the phenomenon under study. This could be useful, for example, to identify contextual anomalies that are linked to specific characteristics of a community. While we are able to achieve good results in several cities, the performance for the Grisons and Ticino cantons are low. These cantons are fairly small and isolated regions. For example, Ticino is highly influenced by the adjacency to Italy this influence is not captured by the model. Two of the main cities in Ticino; Lugano and Belinzona have a very low use of public transport as shown in Fig 5. Another interesting aspect is that the local $R^2$ shows a clustered behavior, with adjacent areas having similar performance. Note that these clusters tend to match administrative boundaries and we can clearly distinguish regions such as Lausanne, Basel and St. Gallen (light blue), central Switzerland (orange) and the Valais (dark blue) in Fig 5. This phenomenon might be linked to the inherent diversity in the communities leaving in the different areas of the country.

**Limitations**

The approach proposed in this paper has few limitations that should be carefully discussed. In details:

**Sample bias**. Even if we showed a fair level of representativeness along different dimensions, the input dataset contains information only on the fraction of population that owns an insurance policy with a specific company. Several segments of the population are left out of the analysis, adding a validity bias in the results, especially for indicators that cover a wider spectrum of the society.

**Spatial granularity mismatch**. Official statistics are available at the level of municipality and only for a subset of the communes, while the insurance customers data provides information at the finer granularity of zip codes. From one side, we have complete knowledge for a subset of the areas, while from the other side, a partial view with a higher coverage. Our analysis is restricted to the intersection between the areas covered by the official statistics and the insurance dataset and, given the spatial units aggregation adopted, it doesn't provides tools to model the heterogeneity of social processes at a micro-level, e.g., neighborhoods in cities.

**Temporal evolution**. In our analysis we currently focus on a static snapshot covering a year of statistics. However, socioeconomic conditions vary over time and in which extent and how fast this change is reflected into the insurance data records is something not explored yet.

**Data availability and privacy**. The current approach is based on the assumption that customers data is available to the researchers to tackle relevant challenges that have a broad social impact. This raises two main issues related to privacy and the compliance to the current legislation especially in the European framework, and the sharing policy. Proprietary data is usually exploited for commercial advantages and profit within the organization, and not available to the broad scientific community. To ground a methodology to model social phenomena on the availability of proprietary data that is not in control of the policy makers raises few concerns on the actual implementation in a real scenario.

# 7 Conclusions

In this paper we proposed 34 different characteristics of individual socio-economic behavior quantifiable through the dataset of anonymized insurance customers, and then evaluated them on the example of Swiss municipalities. We showed that those quantities could be used for estimating economic performance of the regions in the country, as proposed geographical regression models technique demonstrated to perform well on the validation samples for predicting major official statistical quantities for different categories such as Population, Transportation, Work, Space and Territory, Housing and Economy at the level of Swiss municipalities. This approach is applicable in cases when official statistics are not available or they are inconsistent, and the experimental pipeline demonstrated its ability to reach comparable performance to a scenario with complete knowledge. Advantages and disadvantages of local and global spatial regression models have been discussed extensively, highlighting the the potential of insurance customers data to characterize socioeconomic processes embedded in space. In future work, we aim at applying our approach to the modeling of temporal variations, which is especially useful to study processes of urbanization and gentrification. We also aim at developing models for estimating attributes at finer geographical resolutions such as districts or neighborhoods.

## Supporting information

**S1 Appendix.**
(PDF)

**S2 Appendix.**
(PDF)

## Acknowledgments

The authors would like to thank *La Mobili'ere* insurance for partially supporting this research and for providing the anonymized customers dataset.

## Author Contributions

**Conceptualization:** Emanuele Massaro.

**Data curation:** Lorenzo Donadio, Emanuele Massaro.

**Formal analysis:** Rossano Schifanella, Emanuele Massaro.

**Funding acquisition:** Claudia R. Binder, Emanuele Massaro.

**Investigation:** Emanuele Massaro.

**Methodology:** Rossano Schifanella, Emanuele Massaro.

**Project administration:** Emanuele Massaro.

**Resources:** Emanuele Massaro.

**Software:** Lorenzo Donadio, Emanuele Massaro.

**Supervision:** Rossano Schifanella, Claudia R. Binder, Emanuele Massaro.

**Validation:** Emanuele Massaro.

**Visualization:** Emanuele Massaro.

**Writing – original draft:** Lorenzo Donadio, Rossano Schifanella, Emanuele Massaro.

**Writing – review & editing:** Lorenzo Donadio, Rossano Schifanella, Emanuele Massaro.

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
