## [Decision Letter · Decision Letter 0]

21 Jul 2020

PONE-D-20-17957

Leveraging insurance customer data to characterize socioeconomic indicators of Swiss municipalities

PLOS ONE

Dear Dr. Massaro,

Thank you for submitting your manuscript to PLOS ONE. After careful consideration, we feel that it has merit but does not fully meet PLOS ONE’s publication criteria as it currently stands. Therefore, we invite you to submit a revised version of the manuscript that addresses the points raised during the review process.

Both qualified reviewers liked your paper and recommended a minor revision. Reviewer 1 mentioned the time dummy. I think you used only one-year data (2017). Reviewer 2 raised a few important technical and methodological issues. Please try to address their concerns as much as you can.

I also have two minor expositional comments. (1) Some paragraphs are very long and it is not easy for readers. Please consider breaking a long paragraph into two or more paragraphs with each paragraph focusing on only one main point. (2) For tables 6 and 7, to ease readers’ burden, it is better to add a column of variable names, for example, “fraction of owners” for f3. I feel it is troublesome to go back to table 4 to understand what f3 denotes. Also for each model, you can report the R-squared and the stability measure so that readers can have a better understanding of the model fitting.  

We look forward to receiving your revised manuscript.

Kind regards,

Shihe Fu, Ph.D.

Academic Editor

PLOS ONE

Journal Requirements:

2. Our internal editors have looked over your manuscript and determined that it is within the scope of our Cities as Complex Systems Call for Papers. This collection of papers is headed by a team of Guest Editors for PLOS ONE: Marta Gonzalez (University of California, Berkeley) and Diego Rybski (Potsdam Institute for Climate Impact Research).

The Collection will encompass a diverse and interdisciplinary set of research articles applying the principles of complex systems and networks to problems in urban science.  Additional information can be found on our announcement page: https://collections.plos.org/s/cities.

If you would like your manuscript to be considered for this collection, please let us know in your cover letter and we will ensure that your paper is treated as if you were responding to this call. If you would prefer to remove your manuscript from collection consideration, please specify this in the cover letter.

3.We note that [Figure(s) 1, 5 and A1-12] in your submission contain [map/satellite] images which may be copyrighted. All PLOS content is published under the Creative Commons Attribution License (CC BY 4.0), which means that the manuscript, images, and Supporting Information files will be freely available online, and any third party is permitted to access, download, copy, distribute, and use these materials in any way, even commercially, with proper attribution. For these reasons, we cannot publish previously copyrighted maps or satellite images created using proprietary data, such as Google software (Google Maps, Street View, and Earth). For more information, see our copyright guidelines: http://journals.plos.org/plosone/s/licenses-and-copyright.

1.    You may seek permission from the original copyright holder of Figure(s) [1, 5 and A1-12] to publish the content specifically under the CC BY 4.0 license. 

4.Thank you for stating the following financial disclosure:

 [NO].

5.Thank you for stating the following in your Competing Interests section: 

[NO].

6.We note that you have indicated that data from this study are available upon request. PLOS only allows data to be available upon request if there are legal or ethical restrictions on sharing data publicly. For more information on unacceptable data access restrictions, please see http://journals.plos.org/plosone/s/data-availability#loc-unacceptable-data-access-restrictions.

Reviewers' comments:

Reviewer's Responses to Questions

**Comments to the Author**

1. Is the manuscript technically sound, and do the data support the conclusions?

Reviewer #1: Yes

Reviewer #2: Partly

2. Has the statistical analysis been performed appropriately and rigorously? 

Reviewer #1: No

Reviewer #2: Yes

3. Have the authors made all data underlying the findings in their manuscript fully available?

Reviewer #1: Yes

Reviewer #2: Yes

4. Is the manuscript presented in an intelligible fashion and written in standard English?

Reviewer #1: Yes

Reviewer #2: Yes

5. Review Comments to the Author

Reviewer #1: Summary: This paper provides a new method to predict socioeconomic indicators of municipalities using insurance data. Two types of individual insurance data-housing and car insurance-are used. This paper then study adopts two geographical regression models (SLM and GWR) to explore the impact of spatial dependency effects.

Comment：

As the high cost of traditional statistical method to access the demographic data, this paper provides us a new method to estimate statistical data from private insurance data. This paper has done micromesh data analysis work and considered spatial dependency of such work. Some suggestion for revision.

1. The spatial distribution of insurance and Swiss Census data should be compared.

2. Detailed methods in Features selection should be provided.

3. In SLM and GWR model, time dummy should be added in.

4. Why just use SLM model rather than SEM (Spatial error) model? Should provide examination result.

5. Why just use such spatial weight matrix as (5)? Can prepare the different results using different spatial weight matrices?

Reviewer #2: Summary

This paper explores possibilities using insurance customer data as alternative measures of macro social indicators which are conventionally collected through surveys. Using alternative data sources such as google street images and social media posts as measures of social indicators that are otherwise too difficult or too expensive to obtain is not new, but the use of insurance data is novel. The authors have made a careful discussion on the pipeline of prediction algorithm which consists of two steps - first on features selection and then on prediction under a spatial regression framework. This paper is of great interest, but several aspects need to be further clarified. These have been outlined below.

Comments

On pages 4 and 5 in insurance data, the authors said that after matching the insurance data with the survey, 568,426 observations were left from the original dataset of 1,341,328 observations. Could the authors clarify why more than half observations were dropped when matching? Is that due to the lack of geographical identifiers in one of the data sources?

On page 6 in Swiss census data. Could the authors give more information on the representativeness of the census data, for example, how many observations were collected in each municipality? Could the data be interpreted as representative at levels of municipalities or cantons or the nation?

On page 8 in feature selection. Could the authors explain the rationale using LASSO? Why not adaptive lasso or elastic net which may perform better than LASSO?

On page 9 in the spatial lag model. Please clarify the k-nearest neighbors (KNN) scheme? What is the weighting matrix regarding this scheme? Is it based on contiguity measures or kernels? Could the authors show the robustness of results concerning the alternative weighting scheme used?

On page 10. Geographical Weighted Regression should be Geographically Weighted Regression.

On page 10. "In regression analysis, we try to explain the variations of a dependent variable using a suite of uncorrelated and normally distributed independent variables." This statement is not true. In most regression analyses, orthogonality and normality are not preconditions.

On page 9 in the methods. The predictive pipeline proposed by the authors consists of two steps. The first step is feature selection using LASSO, and the second is the prediction of census indicators using selected features. Although it's well-known that the OLS regression after LASSO (so-called OLS post-lasso estimator) performs very well in terms of its statistical properties \\citep{Belloni2013}, the pipeline which uses more complicated models such as SLM or GWR after LASSO may behave differently. Could the authors provide references on the validity of this proposed pipeline?

On page 11 in the results. For each predictor, the selected features range from 2 to 9. It would be interesting to conduct a constrained LASSO selection where the number of features is constrained. Given that the aim of this paper is using insurance claim data as an alternative measure of census-based indicators, it would be more valuable if this could be achieved by a more sparse representation of selected features, say 2 or 3. I would appreciate it if the authors could check if this is the case.

6. PLOS authors have the option to publish the peer review history of their article (what does this mean?). If published, this will include your full peer review and any attached files.

Reviewer #1: No

Reviewer #2: No

---

## [Author Response · Author response to Decision Letter 0]

18 Dec 2020

We would like to thank again the reviewers for appreciating our work and their interesting comments that helped us to improve the work. We have revised the manuscript addressing all the editorial comments. Please find the list of all the changes to the manuscript in our point-by-point response in the attached document.

---

## [Decision Letter · Decision Letter 1]

27 Jan 2021

Leveraging insurance customer data to characterize socioeconomic indicators of Swiss municipalities

PONE-D-20-17957R1

Dear Dr. Massaro,

We’re pleased to inform you that your manuscript has been judged scientifically suitable for publication and will be formally accepted for publication once it meets all outstanding technical requirements.

Kind regards,

Shihe Fu, Ph.D.

Academic Editor

PLOS ONE

Additional Editor Comments (optional):

Reviewers' comments:

Reviewer's Responses to Questions

**Comments to the Author**

1. If the authors have adequately addressed your comments raised in a previous round of review and you feel that this manuscript is now acceptable for publication, you may indicate that here to bypass the “Comments to the Author” section, enter your conflict of interest statement in the “Confidential to Editor” section, and submit your "Accept" recommendation.

Reviewer #1: All comments have been addressed

Reviewer #2: All comments have been addressed

2. Is the manuscript technically sound, and do the data support the conclusions?

Reviewer #1: Partly

Reviewer #2: Yes

3. Has the statistical analysis been performed appropriately and rigorously? 

Reviewer #1: Yes

Reviewer #2: Yes

4. Have the authors made all data underlying the findings in their manuscript fully available?

Reviewer #1: Yes

Reviewer #2: Yes

5. Is the manuscript presented in an intelligible fashion and written in standard English?

Reviewer #1: Yes

Reviewer #2: Yes

6. Review Comments to the Author

Reviewer #1: The author has carefully revised this manuscript according to my 1st review report. After serious review the reviesed version of this manuscript, I think this version can be accepted.

Reviewer #2: This paper explores possibilities using insurance customer data as alternative measures of macro social indicators which are conventionally collected through surveys.

My initial concerns such as validity of merging process of two samples, use of LASSO instead of elastic-net or constrained LASSO, and the pipeline of algorithms proposed for sealing the model selection and parameter estimation have been well addressed by authors and the arguments or revision or future extension are quite convicting and promising. I have no further questions to the authors.

7. PLOS authors have the option to publish the peer review history of their article (what does this mean?). If published, this will include your full peer review and any attached files.

Reviewer #1: No

Reviewer #2: No

---

## [Editor Report · Acceptance letter]

8 Feb 2021

PONE-D-20-17957R1 

Leveraging insurance customer data to characterize socioeconomic indicators of Swiss municipalities 

Dear Dr. Massaro:

I'm pleased to inform you that your manuscript has been deemed suitable for publication in PLOS ONE. Congratulations! Your manuscript is now with our production department. 

Kind regards, 

on behalf of

Dr. Shihe Fu 

Academic Editor

PLOS ONE